# Commercial afforestation can deliver effective climate change mitigation under multiple decarbonisation pathways

Eilidh J. Forster [1], John R. Healey [1], Caren Dymond[2] & David Styles [1,3✉]

Afforestation is an important greenhouse gas (GHG) mitigation strategy but the efficacy of commercial forestry is disputed. Here, we calculate the potential GHG mitigation of a UK national planting strategy of 30,000 ha yr$^{-1}$ from 2020 to 2050, using dynamic life cycle assessment. What-if scenarios vary: conifer-broadleaf composition, harvesting, product breakouts, and decarbonisation of substituted energy and materials, to estimate 100-year GHG mitigation. Here we find forest growth rate is the most important determinant of cumulative mitigation by 2120, irrespective of whether trees are harvested. A national planting strategy of commercial forest could mitigate 1.64 Pg $CO_2$e by 2120 (cumulative), compared with 0.54–1.72 Pg $CO_2$e for planting only conservation forests, depending on species composition. Even after heavy discounting of future product substitution credits based on industrial decarbonisation projections, GHG mitigation from harvested stands typically surpasses unharvested stands. Commercial afforestation can deliver effective GHG mitigation that is robust to future decarbonisation pathways and wood uses.

[1] School of Natural Sciences, Bangor University, Gwynedd, UK. [2] Government of British Columbia, Victoria, BC, Canada. [3] Bernal Institute, School of Engineering, University of Limerick, Limerick, Ireland. ✉email: david.styles@ul.ie

Achieving the ambition of the Paris Agreement to limit global warming to 1.5 °C above pre-industrial levels requires rapid implementation of negative emissions technologies[1], in particular increasing forest carbon sinks[2]. Afforestation on land spared from agriculture[3] is proposed as an integral element of greenhouse gas (GHG) mitigation[4,5], but mitigation efficacy is highly context specific and depends on, inter alia, climate, tree species, management and product substitution. In tropical regions, fast-growing commercial forests managed for wood production have been unfavourably compared to restoration of natural forest[6] but these conclusions are not transferable to a temperate context. There remains a gap in evidence determining the most effective type of temperate forestry for achieving GHG mitigation. The role of sustainable wood harvesting is recognised in GHG mitigation strategies[5,7,8] and could be enhanced through cascading use of wood[9], but recent modelling indicates that future mitigation from product substitution will diminish owing to decarbonisation of substituted materials and energy[10,11]. In coniferous temperate forests, bioenergy harvest breakeven times can range from 30 to 70 years assuming coal is being replaced, depending on the scale of the assessment and whether the stands are damaged[12]. Meanwhile, dynamic life cycle assessment (LCA) studies indicate that it can take up to 100 years for $CO_2$ released from current bioenergy generation to be fully compensated by forest regrowth[13,14].

To fully understand the role of new commercial forests in future GHG mitigation, there remains a need for coherent modelling of GHG mitigation across the four life cycle stages (production, use, cascading use and end-of-life) of hierarchical wood value chains[15] in the context of decarbonised material and energy systems[4]. We fill this gap through comparison of 100-year mitigation achieved by newly planted commercial and conservation (i.e., unharvested) forests in the UK, a country with high afforestation potential[3,16,17]. Although prioritising GHG mitigation in afforestation strategies will involve complex trade-offs with other ecosystem services[18–21], such as biodiversity conservation, the purpose of this paper is not to explore these. Rather, it aims to generate robust evidence on the efficacy of alternative afforestation options for the primary strategic aim of mitigating GHG emissions. We apply dynamic consequential LCA[22] to 33 what-if scenarios representing a range of commercial Sitka spruce (Picea sitchensis) forests with different harvested wood product (HWP) value chains, and conservation forests, across representative growth rates, and in the context of different decarbonisation pathways (Table 1).

Under typical conditions, one hectare of newly planted commercial forest could achieve cumulative GHG mitigation of up to 2.27 Gg $CO_2$e ha$^{-1}$ within a core decarbonisation pathway projected for the UK[4] by 2120—up to 269% more mitigation than delivered by newly planted broadleaf conservation forests, and 17% more than achieved by leaving a newly planted fast-growing conifer forest unharvested. Harvesting reduces long-term terrestrial carbon storage by 61%, but this is more than offset by HWP carbon storage, and concrete and energy substitution—even with near total decarbonisation of the energy sector within the 100-year study horizon. Projected wide deployment of carbon capture and storage (CCS)[4] contributes up to 45% of cumulative 100-year mitigation benefit, but also reduces the energy substitution credits realised by future wood energy. Consequently, the efficacy of mitigation is surprisingly similar between hierarchical and bioenergy uses of wood, and remarkably resilient to CCS deployment assumptions. To contextualise the results in a national planting strategy, we model the UK Committee on Climate Change recommended planting rate of 30,000 ha year$^{-1}$ from 2020 to 2050. If this strategic target is met with commercial forests, it would cumulatively mitigate up to 1.64 Pg $CO_2$e emissions by

2120, compared with 0.54 Pg $CO_2$e for a semi-natural broadleaf conservation forest, or 1.09 Pg $CO_2$e for an equal planting rate of 15,000 ha year$^{-1}$ of each forest type. For UK forests and projected decarbonisation pathways, large-scale commercial afforestation could contribute strongly to GHG mitigation targets.

## Results

**One hundred-year snapshots following a single planting event.** Terrestrial carbon sequestration in semi-natural broadleaf, mixed conifer-broadleaf and fast-growing conifer conservation forests planted in 2020 equates to cumulative mitigation by 2120 of 0.62 (0.41–0.82), 1.30 (0.94–1.64) and 1.94 (1.30–2.58) Gg $CO_2$e ha$^{-1}$, respectively (Fig. 1). Mitigation performance across these conservation forest types is proportionate to yields (Table 1). On average over 100 years, commercial Sitka spruce forests store 61 and 42% less terrestrial carbon than unharvested conifer and mixed conservation forests, respectively, but still 24% more than slow-growing semi-natural broadleaf forests (Fig. 1). However, in addition to terrestrial carbon storage, commercial forests realise ca. 1.7 Gg $CO_2$e ha$^{-1}$ mitigation by HWP carbon storage, fossil fuel substitution, bioenergy carbon capture and storage (BECCS) and concrete substitution in the Core context, at average growth rates (Fig. 1). Relatively small processing emissions of 0.20 (Bioenergy prioritised wood use) to 0.28 Gg $CO_2$e ha$^{-1}$ (Hierarchical wood use) are based on the conservative assumption of a very gradual decline in the emission intensity of processing operations (Table 2 and Supplementary Data 1). Overall, one hectare of yield class (YC) 18 commercial forest planted in 2020 will achieve cumulative GHG mitigation of between 2.0 and 2.27 Gg $CO_2$e by 2120, varying by less than 8 and 10% between wood use strategy and decarbonisation context, respectively. Yield variability extends the mitigation range to 1.36–2.96 Gg $CO_2$e ha$^{-1}$ by 2120. Thus, commercial Sitka spruce forests support between 3% (relative to 100% conifer) and 269% (relative to 100% broadleaf) more GHG mitigation than conservation forests over a 100-year time horizon from establishment.

**National planting programme.** In our scenario of a commercial forest planting programme involving 30,000 ha planted every year from 2020 until 2050 (Fig. 2), only 15% of the 100-year mitigation will be achieved by 2050. A 30-year commercial forest planting strategy could achieve cumulative net mitigation of up to 1.64 Pg $CO_2$e at default yield (Fig. 2), with a range across yields and scenarios of 1.13–2.0 Pg $CO_2$e for YC 14 Hierarchical (Further Ambition) and YC22 Bioenergy (Core), respectively (shown in Supplementary Data 2 and Supplementary Data 3). This compares with mitigation of between 0.54, 1.15 and 1.72 Pg $CO_2$e from equal planting of semi-natural, 50:50 conifer-broadleaf mixed or unharvested conifer forests, respectively (Fig. 2), at default yields (Table 1). A mixed planting strategy of 15,000 ha per year each of commercial forest (YC 18) and semi-natural broadleaf forest (YC 4) could achieve 1.09 Pg $CO_2$e mitigation, 34% lower than commercial forestry alone. For commercial forests established after 2040, cumulative mitigation from regrowth and downstream mitigation does not catch up with cumulative mitigation achieved by unharvested conifer conservation forests by 2120, so that the latter forest type outperforms commercial forests for the 30-year planting programme—despite commercial forests achieving more mitigation across two full rotations (Fig. 1). However, mitigation trajectories (Fig. 2) and 100-year mitigation performance (Fig. 1) indicate that commercial forests would achieve considerably more mitigation than conifer conservation forests beyond our 2120 cut-off. Basing the national forest planting programme entirely on a Bioenergy wood use strategy would achieve annual mitigation of 20.2 Tg $CO_2$e by 2050 from YC 18 Sitka spruce, under a Further Ambition context (Supplementary Data 3), equating to 24% of projected gross

**Table 1 Evaluated forest value chain scenarios, comprising commercial and conservation forests with a mix of conifer and broadleaf species.**

| Forest value chain scenario | Species mix[63] | Yield class[50] (m³ ha⁻¹ year⁻¹ growth) | Harvested thinnings, 1st use | Main harvest, 1st use | HWP 2nd and 3rd uses | Decarbonisation context |
|---|---|---|---|---|---|---|
| Commercial forest-Hierarchical (Core) | 100% Sitka spruce (*Picea sitchensis*) | Default 18, range 12–24[50,64] | 30% boards, 33% short-lived products, 37% electricity | 18% carcassing, 23% boards, 33% short-lived products, 17% electricity, 7% heat | 44% boards, 25% electricity, 14% incineration, 1% landfill, 2% paper production, 14% mulch | Core (see Table 2) |
| Commercial forest-Hierarchical (Further Ambition) | | As above | As above | As above | As above | Further Ambition (see Table 2) |
| Commercial forest-Bioenergy (Core) | 100% Sitka spruce (*Picea sitchensis*) | Default 18, range 12–24[50,64] | 100% electricity | 18% carcassing, 17% short-lived products, 65% electricity | 48% electricity, 19% incineration, 1% landfill, 32% mulch | Core (see Table 2) |
| Commercial forest-Bioenergy (Further Ambition) | | As above | As above | As above | As above | Further Ambition (see Table 2) |
| Semi-natural broadleaf conservation forest | 40% silver birch (*Betula pendula*), 40% rowan (*Sorbus aucuparia*), 20% pedunculated oak (*Quercus robur*) | Default 4, range 2–6[50,65] | NA | NA | NA | NA |
| Mixed conservation forest | 10–30% Sitka spruce, 10–30% Corsican pine (*Pinus nigra* ssp. *laricio*), 5–15% Douglas fir (*Pseudotsuga menziesii*), 10–30% silver birch, 10–30% rowan, 5–15% pedunculated oak | Default 11, range 7.5–14.5[50,65] | NA | NA | NA | NA |
| Unharvested conifer conservation forest | 40% Sitka spruce, 40% Corsican pine, 20% Douglas fir | Default 18, range 12–24[50,64] | NA | NA | NA | NA |

Harvested wood product (HWP) carbon storage, and greenhouse gas emission displacement via product substitution, depend on choice of Hierarchical or Bioenergy wood use strategies, and the degree of decarbonisation in the wider economy—defined by UK Committee on Climate Change. 'Core' and 'Further Ambition' pathways (see Table 2, S1 and Supplementary Data 1).

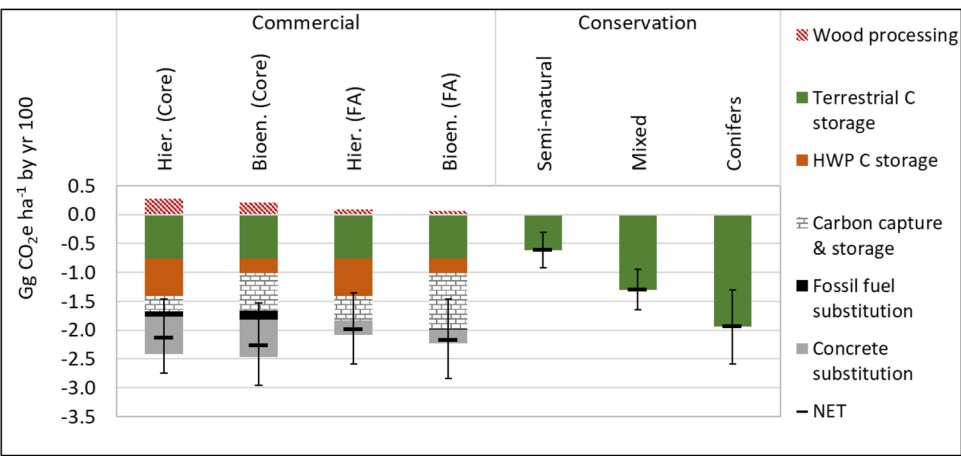

**Fig. 1 Contribution of major processes to cumulative GHG fluxes from one hectare of afforested land (planted in 2020) by year 100, for commercial and conservation forests.** Default yield class 18 ($m^3\ ha^{-1}\ year^{-1}$) for the commercial and unharvested conifer conservation forests, 4 for the semi-natural broadleaf conservation forest and 11 for the mixed conifer-broadleaf conservation forest. Error bars represent full yield ranges presented in Table 1. Commercial forest results are presented for Core and Further Ambition (FA) decarbonisation contexts, and for Hierarchical (Hier.) and Bioenergy (Bioen.) wood use strategies, with differing mitigation contributions from concrete and fossil fuel substitution and carbon storage in harvested wood products (HWP) or long-term geological stores via carbon capture and storage.

anthropogenic GHG emissions in the UK if net zero targets are achieved[4]. Establishing 30,000 ha year⁻¹ of commercial forests could directly supply 3.88 or 14.2 TWh year⁻¹ electricity from thinnings plus first-harvest biomass, from 2070 onwards, based on Hierarchical (Supplementary Data 2) or Bioenergy (Supplementary Data 3) wood use strategies, respectively (end-of-life use of construction wood for bioenergy generates a further 20–43% electricity over the following 50 years—Supplementary Data 2 and Supplementary Data 3).

**Wood use and decarbonisation scenarios.** For a national planting programme of 30,000 ha year⁻¹, more rapid decarbonisation of the wider economy represented in the Further Ambition (vs Core) context could reduce cumulative national mitigation achieved by Hierarchical or Bioenergy wood use in 2120 by 5% or 2%, respectively (Fig. 2). Meanwhile, a Bioenergy wood use strategy rather than Hierarchical wood use strategy could increase cumulative GHG mitigation achieved in 2120 by 5% (Core context) to 8% (Further Ambition context) (Fig. 2). In the Further Ambition context, future process emissions and fossil fuel substitution credits are minor, concrete substitution credits are diminished relative to the Core context and mitigation by BECCS becomes prominent through time (Fig. 2). In fact, BECCS accounts for 33% of cumulative mitigation by 2120 in the Bioenergy wood use strategy, whilst HWP carbon storage accounts for just 5% of cumulative mitigation (Fig. 2), reflecting the fact that 65% of harvested biomass immediately goes to bioelectricity generation (Table 1). Respective figures for the Hierarchical wood use strategy are 16 and 17% for BECCS and HWP carbon storage (Fig. 1). HWP carbon storage of 69 (53–85) Tg C by 2120 in the Hierarchical value chains (Supplementary Data 2) is likely to increase further following subsequent harvests and cascading use of the circa 140 Tg of wood dry matter remaining within the built environment at the end of the studied time horizon. In both Bioenergy and Hierarchical scenarios, the BECCS pool will also continue to grow as biomass and cascaded products reach their final use.

**Sensitivity analysis.** In the national planting programme with Hierarchical wood use, product substitution (concrete) accounts for 0.12–0.32 Pg $CO_2$ (8–20%) of cumulative GHG mitigation by

2120 for Core and Further Ambition contexts (YC 18), respectively. Reducing product substitution by 50% would reduce cumulative GHG mitigation by 5% or 11% for Core or Further Ambition contexts, respectively (Supplementary Data 5). Counterintuitively, concrete substitution makes a greater relative contribution to GHG mitigation in the Further Ambition context than the Core context, because net mitigation is lower and concrete decarbonisation lags energy decarbonisation under the Further Ambition decarbonisation pathway. Alternatively, introducing glulam production (substituting structural steel) alongside development of fibres to substitute oil-derived textile fibres could increase cumulative GHG mitigation achieved in 2120 by 10% or 7% for a national planting programme in a Core or Further Ambition context, respectively (Supplementary Data 5). Potentially, viscose fibre could substitute cotton fibre instead of synthetic fibres, in which case one ha of cotton production could be avoided for every ha of commercial forest planted—which could lead to significant further afforestation and associated GHG mitigation overseas (beyond the scope of this study to quantify). In the pessimistic event of no deployment of CCS technology during the 100-year study time horizon, 100-year mitigation achieved by a commercial forest planting strategy in the UK would be reduced by just 8% (Core) or 6% (Further Ambition) for Bioenergy wood use (Supplementary Data 7) and 6% (Core) or 4% (Further Ambition) for Hierarchical wood use (Supplementary Data 6), respectively. In the absence of CCS, energy and concrete substitution credits are larger than in the base case, counteracting the effects of reduced BECCS and thus reducing the sensitivity of net GHG mitigation to CCS deployment. Across all these alternative product substitutions, commercial forestry remains an effective GHG mitigation strategy, still achieving up to 162% more GHG mitigation than semi-natural conservation forestry in the worst case scenario of reduced concrete substitution.

## Discussion

New evidence presented in this paper emphasises the efficacy of commercial forestry for GHG mitigation, and its resilience to future decarbonisation pathways and wood uses, highlighting the effective and reliable contribution that commercial forestry could make towards the Paris Agreement[1] even with projected

**Table 2 Characterisation of marginal incurred or substituted processes accounted for in dynamic life cycle assessment of forestry value chains, evolving through five time periods over the time horizon considered.**

| Pathway | Energy source or technology | Time period | | | | |
|---|---|---|---|---|---|---|
| | | 2020–2029 | 2030–2039 | 2040–2049 | 2050–2085 | 2085–2120 |
| Core | Marginal industrial heating fuel | Coal | Coal | Coal | 90% coal, 10% biomass | 15% hydrogen, 85% biomass |
| | Marginal electricity generation | Natural gas | Natural gas | Natural gas | Natural gas | Natural gas |
| | CCS deployment on biomass and fossil fuel electricity (%) | 0% | 10% | 20% | 50% | 100% |
| | CCS deployment on biomass and fossil fuel heat (%) | 0% | 0% | 0% | 0% | 45% |
| | CCS deployment on calcination in cement production (%) | NA (no sawn wood production, therefore no concrete substitution) | | | 0% | 10% |
| | CCS deployment on wood panel production emissions—heat plus ammonia/resin production (%) | 0% | 0% | 0% | 0% | 45% |
| | HGV electrification/hydrogen fuel (%) | 0% | 0% | 0% | 13% | 100% |
| Further Ambition | Marginal industrial heating fuel | Coal | Coal | 40% coal, 60% biomass | 15% coal, 85% biomass | 15% hydrogen, 85% biomass |
| | Marginal electricity generation | Natural gas | Natural gas | Natural gas | Solar PV | Solar PV |
| | CCS deployment on biomass and fossil fuel electricity (%) | 0% | 50% | 100% | 100% | 100% |
| | CCS deployment on biomass and fossil fuel heat (%) | 0% | 0% | 45% | 90% | 100% |
| | CCS deployment on calcination in cement production (%) | NA (no sawn wood production, therefore no concrete substitution) | | | 90% | 100% |
| | CCS deployment on wood panel production emissions—heat plus ammonia/resin production (%) | 0% | 0% | 45% | 90% | 100% |
| | HGV electrification/hydrogen fuel (%) | 0% | 0% | 45% | 90% | 100% |

Marginal energy sources and technology deployment reflect Core and Further Ambition decarbonisation pathways projected by the UK Committee on Climate Change[4], long-term marginal electricity supply mixes proposed for life cycle assessment modelling[24] and a road map for decarbonisation of the UK cement industry[66]. See Supplementary Data 1.

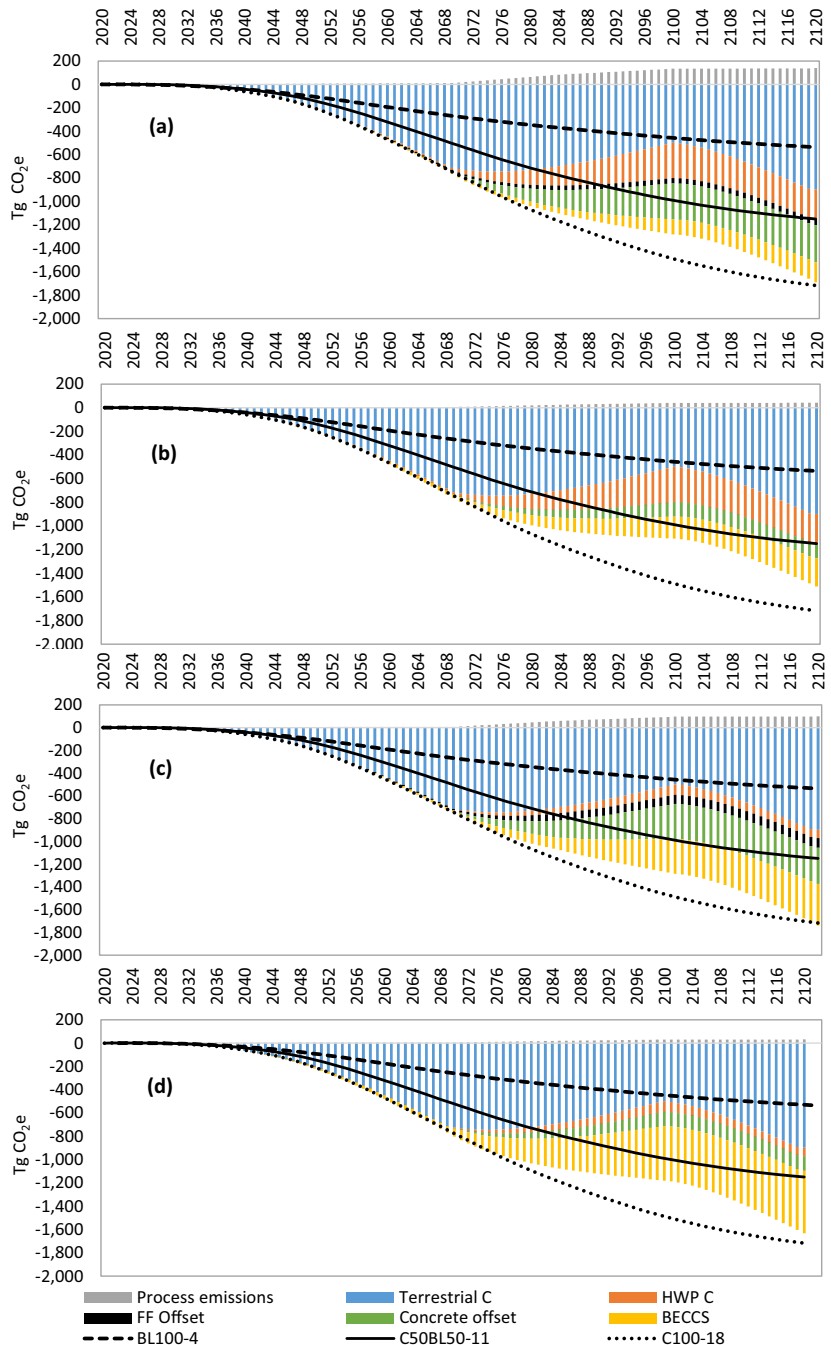

**Fig. 2 Cumulative CO$_2$e mitigation over a 100-year period for commercial forests planted at a rate of 30,000 ha year$^{-1}$ from 2020 to 2050, with a Hierarchical wood use strategy (a, b) or a Bioenergy wood use strategy (c, d), under Core (a, c) or Further Ambition (b, d) decarbonisation contexts.** Commercial Sitka spruce forests are modelled based on default yield class (YC) 18. Contributing factors are shown as stacked bars: process emissions, terrestrial forest carbon storage, harvested wood product (HWP) carbon storage, fossil fuel (FF) substitution (negligible in the Further Ambition context), concrete substitution and bioenergy carbon capture and storage (BECCS). Also shown (lines) are cumulative CO$_2$e mitigation trends for conservation forests planted at the same rate, ranging from semi-natural 100% broadleaf (YC 4), through a 50:50 mixed conifer:broadleaf forest (YC 11) to an unharvested 100% conifer forest (YC 18). These lines represent terrestrial forest carbon storage only, and the C100-18 forest provides a direct comparison of cumulative mitigation that could be achieved by YC 18 conifers if they are not harvested.

decarbonisation of marginal materials and energy that are likely to be substituted by HWPs in the future. These results counter the conclusions of recent studies questioning the climate credentials of commercial forestry[6,10,23]. Our study demonstrates the importance of considering multiple stages of hierarchical wood use, considering, inter alia, consistent time-dependent substitution factors and CCS deployment based on published projections[4,24]. However, we do find that at equivalent growth

rates, commercial forests lag unharvested conservation forests in cumulative GHG mitigation until just before the second harvest cycle (90 years after establishment for typical UK forests).

The magnitude of GHG mitigation achieved by commercial forestry is remarkably resilient to the uses of wood and wider decarbonisation pathways over a 100-year timeframe. Indeed, across all the variables considered, it is clear that tree growth rates have the greatest influence on GHG mitigation, in line with

previous studies focussed primarily on terrestrial carbon storage by forests[25,26]. In this study, the vast majority of product substitution credits attributable to new forest planting are heavily discounted because they arise over 50 years into the future (after first main harvest) when decarbonisation projections indicate much lower emission intensities for substituted energy and materials (Supplementary Data 1). Had current substitution credits been applied throughout, the GHG mitigation achieved by commercial forests would have been considerably greater, and more sensitive to variations in wood use[15,27,28]. Nonetheless, it remains clear that hierarchical use of timber relies on proven technologies to deliver the majority of 100-year mitigation via HWP carbon storage and displacement of mineral construction materials that are unlikely to fully decarbonise for many decades[4,9] (Table 2), maintaining effective mitigation 100 years after first plantings (Fig. 2). Demand for wood is expected to grow as the circular bioeconomy expands and new products are developed (e.g., engineered and modified timber construction products, silvi-chemicals, textiles and biocomposites)[4,26]. Our results show that commercial forests could supply a large share of these future demands whilst delivering comparable or more long-term GHG mitigation compared with conservation forests.

Somewhat surprisingly, both Hierarchical and Bioenergy value chain GHG mitigation was robust to assumptions about CCS deployment. BECCS features prominently in many climate stabilisation scenarios[1,4,5] but depends on commercially unproven technology[29]. Conservatively, our Core decarbonisation pathway involves widespread deployment of BECCS only after 2085, whilst our Further Ambition pathway assumes widespread deployment from 2050 (Table 2)—though with limited effects until the first main harvests from 2070 onwards (Fig. 2). Most climate change mitigation projections indicate a substantial role for dispatchable bioenergy (with CCS) at least up to the end of this century[4,5,17,30]. Global requirements for primary energy from biomass to meet Paris Agreement goals range from circa 100 to 400 EJ year$^{-1}$, and BECCS is expected to play an important role in achieving net zero carbon emissions[5,29–31]. Associated global land requirements have been estimated at between 3 and 7 million km$^2$, with possible negative side effects for biodiversity and food security[5,30,32]. The UK Climate Change Committee recommends installing circa 67 TWh of BECCS electricity generation by 2050[33]. Based on our Bioenergy value chain, over 7 million ha of land would need to be afforested by 2070 in order to fully meet BECCS ambitions exclusively with forest harvests (excluding high-quality sawn wood)—requiring a rate of planting 4.6 times higher, and continuing 20 years longer, than the 30,000 ha year$^{-1}$ over 30 years considered in our study based on UK Committee on Climate Change recommendations[4]. Although a large area, this would bring total UK forest cover to around 42%, close to the European average of 40%[34]. Whilst purpose-grown bioenergy crops could deliver more rapid fossil fuel substitution using less land[33], commercial forests combine the provision of high-quality materials into the circular economy with the delivery of multiple ecosystem services[5,20], including medium-term climate regulation by terrestrial C storage[35] and HWP C storage in the decades before BECCS is anticipated to become widely deployed[4,31] (Table 2). Unlike many purpose-grown bioenergy crops, Sitka spruce can also produce reliable growth on marginal land that could be spared from extensive livestock production in the UK[3].

Commercial afforestation can deliver significantly greater GHG mitigation than semi-natural afforestation largely due to the higher growth rate of conifers. Commercial investment in afforestation is based on the expectation of timber harvest and, to match current timber markets in the UK and the benefit of earlier harvest of faster-growing tree species, this generally leads to the planting of conifer rather than broadleaved species. Thus, faster-growing commercial forests deliver more, and longer-term, GHG mitigation than slower growing semi-natural forests. Depending on future decarbonisation and technology contexts, market demand and environmental risks, high yielding conifer forests could be harvested[36] for hierarchical wood use or for bioenergy (with CCS), or left in the ground to maximise terrestrial C storage. The latter option can be chosen late in the forestry cycle in response to relevant market, environmental and/or policy signals. Commercial forestry therefore provides a robust strategy for GHG mitigation across a wide range of future contexts, at least in temperate zones, and merits a prominent role in climate policies, including nationally determined contributions under the Paris Agreement, and the forthcoming EU Forest Strategy[36].

Our results pertain to regions with commercial forest rotations of 50 years (or less). In regions (e.g., boreal) with slower tree growth rates and longer forestry cycles, leaving wood unharvested could be a better mitigation option over the next century[5,18,36]. We applied a 100-year cut-off to our analysis to maintain a realistic time horizon for climate policy decision making, and to avoid deeply uncertain marginal technologies beyond this time horizon. A longer time horizon is likely to favour harvested commercial forests owing to saturating carbon sequestration in conservation forests (Fig. 2). There is some uncertainty about whether old forests may in fact continue to act as significant carbon sinks[19,37,38]. Unexpected losses from pests, pathogens, fires, windthrow and drought-caused mortality pose an increasing risk to GHG mitigation from affected areas, but such losses will not necessarily affect commercial forests more than conservation forests. In fact, active management and periodic harvesting of commercial forests can reduce fire risks[39,40] and safeguard carbon stocks within HWPs whilst climate change increases environmental risks to old forests.

Whilst results in this paper demonstrate the comparatively high long-term GHG mitigation efficacy of commercial forests compared with unharvested conservation forests, it is imperative to note that commercial and conservation forests deliver different suites of important ecosystem services and should be considered complementary and not conflicting land uses[5]. Rather than limiting the area of commercial forests in favour of semi-natural forests[6], more emphasis should be placed on rapid deployment of both forest types, and on the cascading use of harvested wood to derive maximum economic and mitigation value. Although our modelled commercial forestry system is 100% Sitka spruce, single species forests are not the only commercial option, and there is increasing support for mixed-species conifer forests (or conifer-broadleaf mixtures), under sustainable forest management to deliver both a high yield of wood products and other ecosystem services[20,41]. Diversification of commercial forestry could also enhance resilience to pests, diseases and climate change effects[42–44]. Nabuurs et al.[8] argue for policy changes to encourage such 'climate-smart forestry', claiming it could deliver an extra 441 Mt CO$_2$e year$^{-1}$ mitigation across Europe by 2050. National afforestation strategies should promote a portfolio of commercial, conservation and agroforestry systems tailored to local conditions and adapted to climate change[5,45], in order to deliver the range of ecosystem services required by society, in addition to GHG mitigation[46]. To achieve this there is a strong case to increase the 30,000 ha year$^{-1}$ UK afforestation area target of the Committee on Climate Change, which is very modest in the context of international ambitions for increased tree canopy cover, such as the 0.9 billion hectares identified by Bastin et al.[47], or indeed the EU Forest Strategy objective to plant 3 billion trees by 2030[36]. Successful forest policy will also need to integrate social outcomes, and engage local stakeholders in strategic planning[41].

We conclude that fast-growing tree species should be a major component of national afforestation strategies in order to achieve

ambitious GHG mitigation targets. Commercial investment will favour such high production forests of species that produce timber matching market demand. The GHG mitigation potential of commercial forestry is robust against future decarbonisation scenarios, and will continue to deliver mitigation well beyond the 100-year timeframe considered in this study, when conservation systems have reached their peak capacity. Furthermore, commercial forestry can deliver similar long-term mitigation when wood is used in either hierarchical or bioenergy value chains, potentially contributing to future circular bioeconomy, energy security and carbon removal objectives as required at the time of harvest. However, it takes time to establish large areas of new forest. In addition to the time lag between forest establishment and GHG mitigation, especially pronounced when considering mitigation from commercial forest hierarchical value chains, there is a time lag in implementation of national afforestation policies owing to constrained planting rates[48]. Whilst sustainable intensification[3] and curtailed demand for livestock products[5,31,49] are projected to free up substantial land for afforestation in the UK and elsewhere over coming decades, our results reveal the urgency of commencing a programme of large-scale tree planting if afforestation is to significantly contribute to Paris Agreement targets for the second half of this century[1,48].

## Methods

**Aim and scenarios**. This study seeks to determine the most effective afforestation strategy for achieving climate change mitigation in a temperate region. Specifically, it aims to generate new evidence on the comparative long-term (100-year) GHG mitigation efficacy resulting from the establishment of commercial forests and conservation forests on grassland in the UK (baseline). We modelled 33 scenarios representing common temperate commercial and conservation forestry typologies, tree growth rates ($m^3\ ha^{-1}\ year^{-1}$: 'yield class') and HWP value chains (Table 1). Rationale for scenario development commensurate with study objectives and conclusions is fully elaborated in S1, and summarised below.

Commercial forestry was based on conifer (Sitka spruce) forests, with a conservative default YC of 18 $m^3\ ha^{-1}\ year^{-1}$ (range 12–24)[50], with intermediate forest thinning in year 21 of each rotation when 36% of the harvestable material is removed in HWP. Clear-fell harvest is implemented in year 50 (a conservative average for this species in UK conditions), followed by immediate replanting, enabling a second clear-fell in year 100. Non-merchantable biomass was assumed to be left to decay on site. Two predominant types of wood value chain were modelled, representing different wood use strategies: (i) a Hierarchical wood use strategy, with wood flow based on UK statistics and data from mills that maximise the value of wood products, and embodying the waste hierarchy[51] (Fig. 3), i.e., wood is subject to cascading use via a prioritised order: durable wood product manufacturing, extending service life times, re-use, recycling, bioenergy generation and disposal[52]; (ii) a Bioenergy wood use strategy, in which high-value sawn wood is used in the construction sector (as in the Hierarchical wood use strategy), but all other wood, and end-of-life construction wood, is used for electricity generation to contribute to bioenergy targets[4]. Dynamic assessment of incurred and avoided emissions during future wood harvest and processing operations, construction material and fossil fuel substitution, and BECCS deployment, was applied based on decadal progression of marginal technologies in line with two sets of projections made by the UK Committee on Climate Change: 'Core' and 'Further Ambition' pathways[4]—detailed in Table 2. These pathways involve differing rates of fossil fuel replacement by low carbon and renewable technologies, and differing rates of CCS implementation across industry sectors, representing different future contexts in which forestry value chains could exist. Note that fossil fuels remain the *marginal* source of dispatchable energy generation well into the future, especially for industrial heat, despite renewable energy technologies predominating[24]. Very few studies to date consider the timing of GHG emissions[41] and avoided GHG emissions in this way.

Forests may be established for varying conservation objectives, from habitat provisioning to terrestrial carbon storage[5,7]. We represented a diverse range of possible conservation (unharvested) forest types based on different ratios of conifer (Sitka spruce, Corsican pine, Douglas fir) to broadleaf (silver birch, rowan, oak) trees (Table 1). The 100%-conifer conservation forest (YC 18) represents carbon sequestration prioritisation and provides an unharvested comparator for the equal-YC commercial forests. Meanwhile, the semi-natural 100%-broadleaf forest represents prioritisation of habitat provisioning, and delivers much lower rates of carbon sequestration owing to slow growth (YC 4)—representative of broadleaf forests grown on lower grade land in UK[50]. Among a range of intermediate conservation forests, the default conservation comparator, delivering a mix of carbon sequestration and habitat provisioning, was a 50:50 conifer:broadleaf mix

with YC 11. Yield classes for 100% Sitka spruce (18) and 100% broadleaved (4) are for the same site conditions.

Scenarios were analysed at two scales: 1 ha planted in 2020, and a national scenario of 30,000 ha $year^{-1}$ planted from 2020 to 2050 as per UK climate recommendations[4,17] on 900,000 ha of lower-quality grassland that could be taken out of livestock production through intensification[3] and/or reduced market demand for livestock products[4,49]. We assume that social and political change will lead to sufficient land being available, therefore, avoided GHG emissions from reduced livestock production or additional emissions from livestock production displacement are not considered. Analyses were cut off at 2120 in both scenarios, representing a maximum time horizon for climate policy. Climate change impacts on forest productivity were not included due to the already high complexity of the scenarios.

**Life cycle assessment goal and scope**. We applied dynamic consequential LCA[22] to evaluate the GHG emissions balance of afforesting one hectare of grassland in the UK, accounting for terrestrial (soil and biomass) carbon storage, HWP carbon storage, substitution of materials and fossil fuels and long-term sequestration of biogenic carbon via future deployment of BECCS, over a 100-year period. Expanded LCA boundaries (Fig. 4) encompassed: (i) land use change due to afforestation on spared agricultural grassland; (ii) forest establishment; (iii) forest growth; (iv) forestry operations; (v) debarking; (vi) sawmilling (including drying, planning and chemical treatment); (vii) wood panel production; (viii) paper and paperboard production; (ix) bioenergy generation, including BECCS; (x) credits for avoided use of fossil fuels (substituted energy generation and mineral construction material production); (xi) carbon storage in HWPs related to 'decay' (retiral) functions[53] and (xii) recycling and disposal of retired HWPs, including via (ix). The production and transport of all material and energy inputs were accounted for, as were the construction or manufacture of infrastructure and capital equipment. Full life cycle inventories are provided in Supplementary Data 2 and Supplementary Data 3, with an example for the Hierarchical wood use value chain in Supplementary Table 1 (S1). Material flows were derived using UK data from a combination of forest carbon modelling[54], harvest data from over 2,000 ha of commercially managed forests, data form a commercial sawmill that maximises sawn wood output, national recycling data[55] and timber-use statistics[56]—elaborated in S1. Given the focus of this paper on GHG mitigation, only the global warming potential ($GWP_{100}$) impact category was evaluated, expressed as kg $CO_2$e. Limitations are explored in the Discussion.

**Terrestrial carbon**. Forest growth, decay and harvest volumes were calculated using the Carbon Budget Model for the Canadian Forest Sector (CBM-CFS3)[54], because it complies with the Intergovernmental Panel on Climate Change guidance[53], was developed for similar climate conditions to the UK and provides default volume to biomass equations for Sitka spruce and Douglas fir (which are native to Canada), and genus equations for the other species in this study. It was parameterised using best-fit yield tables from Forest Yield, the standard yield model for forest management in the UK[50]. For conservation forests, aggregate group YCs were calculated based on weighted mean YCs for the species mixes described in Table 1. CBM-CFS3 outputs include annual soil and biomass carbon stocks and flows of carbon in harvested timber. Harvested carbon from CBM-CFS3 was converted to merchantable volume based on a 49.95% carbon content in wood dry matter[57]. Detailed material flows are presented in Supplementary Data 2 and Fig. 3 for the Hierarchical wood use strategy, and in Supplementary Data 3 for the Bioenergy wood use strategy.

**Process emissions and carbon storage**. All emissions associated with the production and transport of forestry inputs and all primary and secondary wood processing phases were extracted from Ecoinvent v.3.5[58] using OpenLCA v1.7.4 and scaled in Microsoft Excel using the HWP material flow (Supplementary Data 2 and Supplementary Data 3). Emissions from landfill disposal were calculated according to the IPCC First Order Decay method[53]. Retirement rates of HWP were calculated according to IPCC methods[59], with modified decay factors[60]. As products retire from the HWP carbon pool, they are recycled or disposed of (by incineration or landfill) in proportions calculated according to UK recycling statsistics[55] (Fig. 3). Retired tertiary products (i.e., products that have already been recycled once) are conservatively assumed to be disposed of. Horticultural mulch is assumed to decay at a rate similar to composted municipal solid waste[61]. Wood fuel is not included in the HWP carbon pools owing to rapid oxidation; return of this biogenic carbon to the atmosphere from the forest ecosystem is represented in the deficit between ecosystem carbon loss at the point of harvest, HWP carbon storage gain and loss from recycled materials.

**Substitution credits**. Fuel-to-energy conversion factors (for coal, natural gas and wood chips) were taken from Ecoinvent unit processes[58] to calculate fossil fuel substitution by dedicated biomass energy generation and incineration with energy recovery for wood waste. Emission avoidance through substitution of mineral construction materials was estimated by translating the final mass of construction timber (150 tonnes at 20% moisture per ha thinned forest) into an area of timber-framed wall using industry standard design: 0.0175 $m^3$ of timber per 1 $m^2$ wall

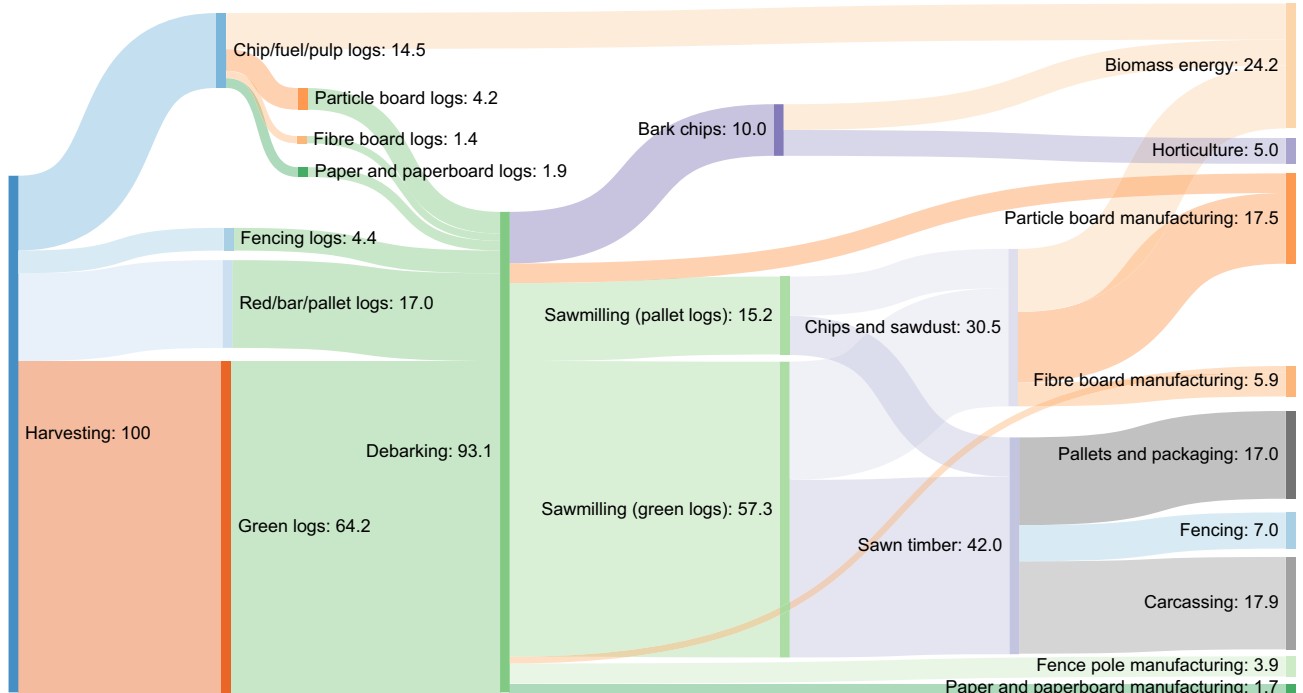

**Fig. 3 Biogenic carbon material flow from wood harvest going to Hierarchical use.** Units are percentages of original harvest calculated in the CBM model[54]. 'Red' and 'Green' refer to a quality threshold of acceptable straightness, taper and knots in a log accepted by sawmills, with 'Green' being the higher quality. Carcassing refers to construction grade sawn wood. Data sources include wood harvest volumes from the CBM-CFS3 model, sawmill product breakouts, UK statistics on wood waste and harvested wood product decay functions, elaborated in relevant sections of Methods and in S1.

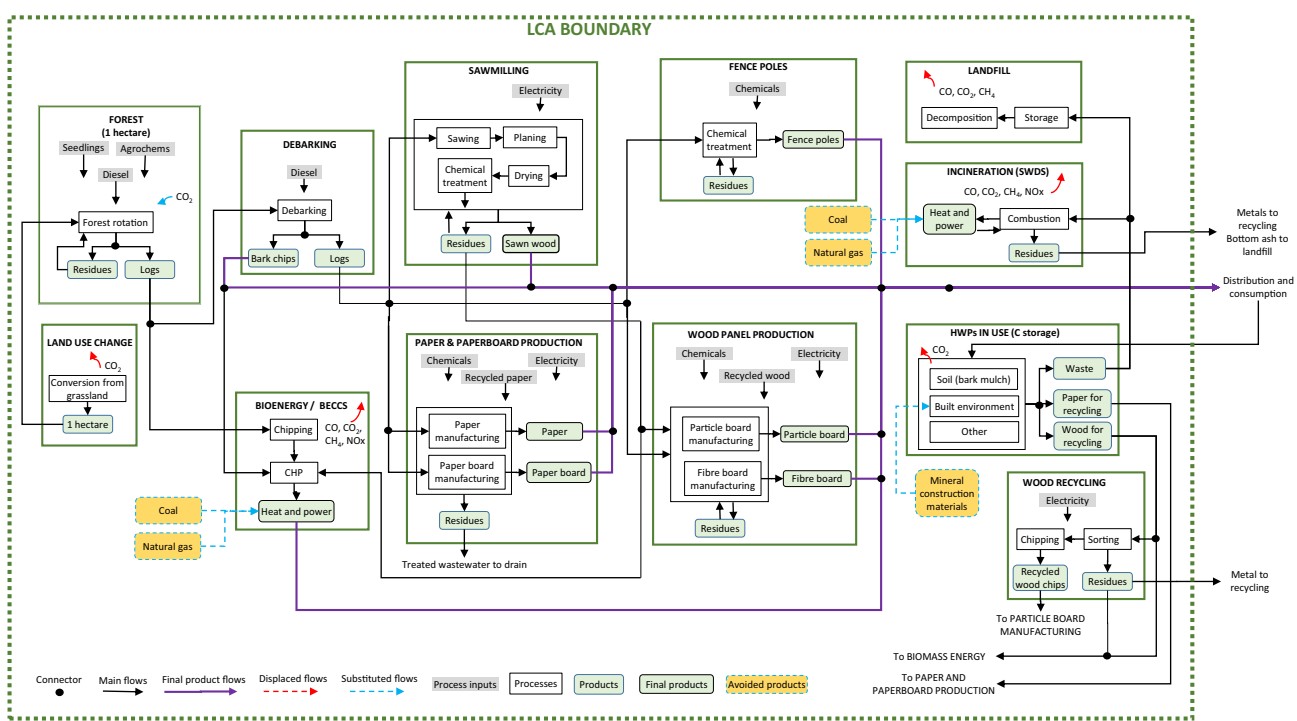

**Fig. 4 Main processes and inputs accounted for during primary, secondary and end-of-life uses of harvested wood products (HWP) within consequential life cycle assessment boundaries.** Main processes arise during each stage of the value chain, including forest establishment (and associated land use change from grassland); wood processing (debarking, milling, wood panel and paper production from primary and secondary wood flows); storage of carbon in HWP (primary and secondary wood products); recycling and disposal of HWP (primary and secondary wood products), including incineration (with energy recovery) and landfill; bioenergy generation (with bioenergy carbon capture and storage (BECCS) in later years); substitution of fossil fuels for energy generation; substitution of mineral construction materials.

**Table 3 Summary of product substitution sensitivity tests (alternative scenarios) performed on main results (base case scenario).**

| Sensitivity test | Base case scenario | | Alternative scenario | |
|---|---|---|---|---|
| | Product | Substituted product | Product | Substituted product |
| Reduced substitution | Carcassing (100% product substitution) | Concrete | Carcassing (50% product substitution, 50% no substitution) | Concrete |
| Increased substitution | Pallets and packaging | None | Glulam beam | Structural steel beam |
| | Wood panels | None | Viscose fibre | Fibres from oil derivates (PET, recycled PET, polyamides)[67] |
| Zero CCS deployment | BECCS | Fossil fuels (with CCS) | Bioenergy (no CCS) | Fossil fuels (no CCS) |
| | Carcassing | Concrete (with CCS) | Carcassing | Concrete (no CCS) |
| | Wood panels (with CCS) | None | Wood panels (no CCS) | None |

'Carcassing' refers to sawn structural timber.

(BRE IMPACT database[62] accessed via eToolLCD® software—Supplementary Table 1). Then, 1 m² of timber frame wall replaces 1 m² of single skin, 140 mm concrete block and mortar (sand:cement ratio 10:3) wall with 10-mm jointing in typical UK house construction. Avoided emissions were then calculated using emission factors from Ecoinvent for the manufacture of concrete blocks, sand and cement. Projected decarbonisation of the cement sector is represented in Core and Further Ambition contexts (Table 2). Assumptions about substitution are elaborated in S1, and reflect marginal changes associated with additional wood supply (e.g., an increase in timber frame construction, hence substitution of mineral construction materials). CCS deployment is gradually increased on a decadal basis, at a faster rate for the Further Ambition context, and applied equally to marginal (substituted) fossil fuel and wood electricity generation (Table 2 and S1).

**Sensitivity analysis**. Recent papers have highlighted the uncertain assumptions inherent in wood product substitution estimates[15,27]. This study presents a methodological approach that addresses key limitations by considering different value chains and dynamic product counterfactuals under different decarbonisation pathways. To further explore the complexity and uncertainty of future wood value chains, we modelled alternative product substitution assumptions to assess the sensitivity of model results to product substitution. Details of these are provided in S1 and Supplementary Data 5, summarised in Table 3. We also modelled an alternative decarbonisation scenario in which there is no deployment of CCS during the study timeframe (Supplementary Data 6 and Supplementary Data 7), to evaluate the dependency of results on this technology.

**Reporting summary**. Further information on research design is available in the Nature Research Reporting Summary linked to this article.

## Data availability
The authors declare that the data supporting the findings of this study are available within the paper and its supplementary information and data files: S1 for elaboration of scenarios and assumptions, Supplementary Data 1 for projected energy and product emission intensities, Supplementary Data 2 and Supplementary Data 3 for Hierarchical and Bioenergy wood use strategies, respectively, Supplementary Data 4 for graphs, Supplementary Data 5 for sensitivity analyses on substitution effects, Supplementary Data 6 and Supplementary Data 7 for full Hierarchical and Bioenergy inventory results, respectively, without future deployment of carbon capture and storage (sensitivity analyses). Background data were generated using the publicly available CBM-CFS3 model and extracted from the Ecoinvent v.3.5 database. All subsequent calculations were undertaken using standard MS Excel functions.

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

## Acknowledgements
This work was funded by the UK Natural Environment Research Council (NERC) Envision Doctoral Training Program. Additional CASE studentship funding was provided by Woodknowledge Wales (Ffarm Moelyci, Tregarth, UK) and Coed Cymru (The Forest Hub, Unit 6, Machynlleth, Powys, UK). The authors are grateful for expert advice received from Gary Newman (Woodknowledge Wales) and Gareth Davies (Coed Cymru) and for expert advice and data supplied by Stephen Ramage (Gresham House) and Rob MacKenna (James Jones and Sons Ltd.).

## Author contributions
E. J. F. undertook data collection and analysis, and drafted the manuscript. D. S. and J. R. H. informed study design and edited the manuscript. C. D. assisted with CBM and wood product modelling and edited the manuscript.

## Competing interests
The authors declare no competing interests.
