## [Peer Review File · Nature Communications]

REVIEWER COMMENTS

Reviewer #1 (Remarks to the Author):

This study estimates climate change mitigation potential of increasing commercial afforestation vs conservation forests in the UK, based on carbon storage of tree growth and substitution of fossil products with wood-based products. The study also accounts for potential decarbonisation of fossil-based products that wood would substitute. The LCA boundaries covered from land use to disposal, including elements of circularity. This is an ambitious and challenging exercise, especially considering the number of scenarios explored (various types of forest types and future timelines). It is also tied to local policy targets.

I don't have a strong expertise with this method, so I will limit my considerations to my general understanding of it, as well as to the general framing of the study and the implications drawn. These kind of counterfactual scenario exercises are important in contributing to informing climate change mitigation. However, I would wish the authors to be much more cautious in regard to drawing general implications for land use planning. This accounts to various reasons.

1. The initial framing in the introduction should be strengthened. The text reads that 'recent papers have suggested that commercial forests are less effective compared with (semi-)natural forests, owing to periodic removal of terrestrial biomass carbon stores through harvesting and albedo effects'. I think it is important to note that those papers mainly refer to planting trees in the tropics and subtropics, where most forest-restoration commitments are found and where the environmental and socio-economic premises are different. I find it confusing to frame the paper so vaguely, in the sense that climate change potential is context-specific. For instance, due to slow growth in boreal forests, carbon storage potential of commercial vs conservation forests is debated. Instead, it would be beneficial to mention, in the introduction, studies about climate mitigation potential in temperate forests (even if only at land use); and to clarify that results from this exercise cannot be generalized to other contexts.

2. Even with my limited knowledge of the method, I know that a large amount of assumptions have to be made in LCA in general, and dynamic LCA in particular. Challenges include data scarcity/uncertainty on, for instance, substitution factors or the soil carbon emissions; or unforeseen changes in the market, including rebound effects due to higher consumption levels. These assumptions, of course, have a large influence on the results. Such limitations are not really presented or discussed in text, with the exception of decarbonisation of alternative products and, (very briefly mentioned) changes in markets/policies.

3. Commercial forestry presents important trade-offs with biodiversity and other ecosystem services. While this is only quickly mentioned in the discussion, it represents a central point and it calls for cautiousness. Carbon storage needs to be balanced against other objectives, including conservation. Land use management decisions cannot be solely informed by estimates of climate change mitigation. Local and regional strategies should be thus evaluated based on the context.

Reviewer #2 (Remarks to the Author):

In the article titled Commercial forestry delivers effective mitigation under multiple decarbonization pathways, the authors present an analysis to demonstrate that commercial forests have a negative higher impact in greenhouse than conservation forests in a period of 100 years. The paper is really interesting, but I have some comments trying to improve it, but also come concerns.

What kind of trees are the ones to be planted?

What kinds of trees serve as a basis for comparison?

There are many acronyms that are not defined.

20505 appears in some parts of the text. Is that correct? I think it refers to the year 2050.

System definition and scope

I read four pages to know that it is a abeto plantation. I suggest that this be mentioned from the beginning.

There are many acronyms that are not defined.

There is no justification for analyzing the scenarios described in Table 1.

It is mentioned that only global warming indices are analyzed and it is understandable, but I am concerned that they do not integrate indices associated with the exhaustion and use of water, since without this resource, the project would not be possible.

Where water is analyzed in the processes defined in Figure 1 related to the limits of the system?.

It is often mentioned that statistics from Canada and the UK are used. Is that acceptable, are there the same conditions?

Results and discussion

Many terms are used that are scientifically trivial, such as requiring a long period of time and such. Better to say the exact number in years.

Conifers are frequently referred to, but which ones are not indicated. I recommend referring to plants and trees by their scientific name.

Figures are riddled with acronyms that are not defined in the text and looking through the tables is quite tiring.

Some paragraphs are excessively long, such as the one on page 15, which contains more than one page and is 401 words long. The central idea is lost.

How risk was integrated in the analysis for different scenarios, such as floods or fires. How do those events affect the system?

Conclusion. I consider that there is a great information analysis and without a doubt that the work carried out is quite a bit by the authors, unfortunately the structure for the article detracts from its quality. There are many tables and acronyms that are not defined in the text, a trivial language is used, often referring to "long time", "conifers", which leaves many doubts.

Under such circumstances, I recommend major changes for the authors, considering that this work contains valuable information, of application and of national interest.

Response to reviewers' comments

	Comment	Response
R1	This study estimates climate change mitigation potential of increasing commercial afforestation vs conservation forests in the UK, based on carbon storage of tree growth and substitution of fossil products with wood-based products. The study also accounts for potential decarbonisation of fossil-based products that wood would substitute. The LCA boundaries covered from land use to disposal, including elements of circularity. This is an ambitious and challenging exercise, especially considering the number of scenarios explored (various types of forest types and future timelines). It is also tied to local policy targets. I don't have a strong expertise with this method, so I will limit my considerations to my general understanding of it, as well as to the general framing of the study and the implications drawn. These kind of counterfactual scenario exercises are important in contributing to informing climate change mitigation. However, I would wish the authors to be much more cautious in regard to drawing general implications for land use planning. This accounts to various reasons.	We are grateful to the reviewer for this well-considered advice. We have extensively rewritten the relevant sections of the manuscript to better acknowledge the key caveats and limitations and to avoid stating unwarranted implications for land use planning. To aid readers in deciding on the implications of the results, we have redesigned the figures to show yield-related uncertainty ranges.
	1. The initial framing in the introduction should be strengthened. The text reads that 'recent papers have suggested that commercial forests are less effective compared with (semi-)natural forests, owing to periodic removal of terrestrial biomass carbon stores through harvesting and albedo effects'. I think it is important to note that those papers mainly refer to planting trees in the tropics and subtropics, where most forest-restoration commitments are found and where the environmental and socio-economic premises are different. I find it confusing to frame the paper so vaguely, in the sense that climate change potential is context-specific. For instance, due to slow growth in boreal forests, carbon storage potential of commercial vs conservation forests is debated. Instead, it would be beneficial to mention, in the introduction, studies about	We thank the reviewer for this important point. We have therefore rewritten the introduction to emphasise the context-specific nature of mitigation, and to highlight that we focus on one context: temperate forests in the UK. Within the limited word count limit of the Introduction, we have expanded the framing by editing down the headline results, and have focused on the need to integrate terrestrial carbon storage with downstream mitigation via harvested wood use in order to draw robust conclusions on long-term mitigation from forests.

	climate mitigation potential in temperate forests (even if only at land use); and to clarify that results from this exercise cannot be generalized to other contexts.	
	2. Even with my limited knowledge of the method, I know that a large amount of assumptions have to be made in LCA in general, and dynamic LCA in particular. Challenges include data scarcity/uncertainty on, for instance, substitution factors or the soil carbon emissions; or unforeseen changes in the market, including rebound effects due to higher consumption levels. These assumptions, of course, have a large influence on the results. Such limitations are not really presented or discussed in text, with the exception of decarbonisation of alternative products and, (very briefly mentioned) changes in markets/policies.	The new S1 section that we have added provides detailed information on the assumptions and scope of the study, and we have revised the Methods section (within the constrained word count limit) to flow more logically. As now elaborated in S1, we employed a conservative and iterative approach to derive scenarios, which reflect a range of future possibilities. These possibilities, along with yield variations, enabled us to carry out extensive sensitivity analyses, which are crucial for researching the effects of key uncertainties and assumptions in this kind of analysis. BECCS is by some margin the most uncertain component, which we discuss extensively. We also now emphasise the surprisingly small effect of harvesting and different wood uses on cumulative mitigation achieved by year 100, which shows that apparently important assumptions actually have a small influence on final results. We also note that Fig. 1 does display the system boundaries, and full life cycle inventories are made available in the SI files for those wishing to dig deeper into our methods.
	3. Commercial forestry presents important trade-offs with biodiversity and other ecosystem services. While this is only quickly mentioned in the discussion, it represents a central point and it calls for cautiousness. Carbon storage needs to be balanced against other objectives, including conservation. Land use management decisions cannot be solely informed by estimates of climate change mitigation. Local and regional strategies should be thus evaluated based on the context.	We welcome this important point made by the reviewer, which we fully agree with. Therefore, we now discuss this in appropriate detail (within the word limit) in the final section of the manuscript.
R2	In the article titled Commercial forestry delivers effective mitigation under multiple decarbonization pathways, the authors present an analysis to demonstrate that commercial forests have a negative higher	The tree species used in the analysis are now clearly identified by their scientific names in Table 1 and the Methods section. Sitka spruce, selected as the commercially dominant tree species in the

impact in greenhouse than conservation forests in a period of 100 years. The paper is really interesting, but I have some comments trying to improve it, but also come concerns. What kind of trees are the ones to be planted?	UK and environmentally-similar areas of Western Europe, is now explicitly referred to throughout the manuscript as the study's commercial forestry species.
What kinds of trees serve as a basis for comparison?	These are now fully described in Table 1, with elaboration added in S1.
There are many acronyms that are not defined.	Acronyms have been removed where possible, and elaborated at first use where unavoidable.
20505 appears in some parts of the text. Is that correct? I think it refers to the year 2050.	We did a thorough search but could not find this.
System definition and scope I read four pages to know that it is a abeto plantation. I suggest that this be mentioned from the beginning.	Sitka spruce (as the specific abeto (conifer) species used in the study) is now explicitly mentioned at the first appropriate opportunity in the Introduction.
There is no justification for analyzing the scenarios described in Table 1.	We have now added a whole new supplementary section, S1, a Word file in which we elaborate on scenario development in detail.
It is mentioned that only global warming indices are analyzed and it is understandable, but I am concerned that they do not integrate indices associated with the exhaustion and use of water, since without this resource, the project would not be possible.	As commented by reviewer 1, we acknowledge that it is important for us to place the study more firmly into the UK upland context of all its scenarios. With a mean annual rainfall of > 1000 mm and no months with < 50 mm mean monthly rainfall, combined with moderate summer temperatures, in this large area available for afforestation tree growth is not limited by water availability now or under any major future climate predictions.
Where water is analyzed in the processes defined in Figure 1 related to the limits of the system?	Potable water is an input to processing operations that carries a small carbon footprint, hence emissions arising from it use are considered within system boundaries, as depicted in Fig.1.
It is often mentioned that statistics from Canada and the UK are used. Is that acceptable, are there the same conditions?	The Carbon Budget Model for the Canadian Forest Sector (CBM-CFS3) was used, but was fully parametrised with UK forest growth data. I.e. no statistics from Canada are used. The CBM-CFS3 model is used to model forest carbon budgets in many countries. It is used, for example, to produce data for Ireland's national GHG inventory report. We therefore consider use of this model, parameterised fully with

		UK statistics, to be appropriate.
	Results and discussion Many terms are used that are scientifically trivial, such as requiring a long period of time and such. Better to say the exact number in years.	We have carried out an extensive rewriting of the results & discussion sections to address the totality of reviewer comments and have been careful to follow this specific advice about terminology.
	Conifers are frequently referred to, but which ones are not indicated. I recommend referring to plants and trees by their scientific name.	The scientific names of the study species are now included in Table 1 and in the text.
	Figures are riddled with acronyms that are not defined in the text and looking through the tables is quite tiring.	Almost all acronyms have now been removed from the figures, and where they have to be retained they are carefully defined.
	Some paragraphs are excessively long, such as the one on page 15, which contains more than one page and is 401 words long. The central idea is lost.	Paragraphs have been reduced in length during rewriting so that each one focuses on a single central idea.
	How risk was integrated in the analysis for different scenarios, such as floods or fires. How do those events affect the system?	Risk, due to such events, is explicitly mentioned in the discussion.
	Conclusion. I consider that there is a great information analysis and without a doubt that the work carried out is quite a bit by the authors, unfortunately the structure for the article detracts from its quality. There are many tables and acronyms that are not defined in the text, a trivial language is used, often referring to "long time", "conifers", which leaves many doubts. Under such circumstances, I recommend major changes for the authors, considering that this work contains valuable information, of application and of national interest.	We have fully understood and reflected on this advice and have carried out a substantial revision of the manuscript in order to correct these deficiencies of structure, explanation and presentation, including the avoidance of acronyms or their full explanation, use of unspecific language.
	See policies and forms required for resubmission in email from editor	

REVIEWER COMMENTS

Reviewer #1 (Remarks to the Author):

I think the authors still need to sharpen the key message of this manuscript, as the entire exercise is to provide insights for national decision-making. Please consider the following points.

1. Should the title include 'in the UK'? Especially in consideration of what pointed out by myself and R2 in the previous revision round.
2. Line 44 'recognised in mitigation strategies' -> what mitigation strategies?
3. Line 48 'replaced'-> I think this should be 'replace'
4. Could you quickly explain, in the intro and discussion, how the number 30,000ha/yr was selected? I understand that it is a target/recommendation suggested by the UK Committee on Climate Change, but this is not so evident to any reader.
5. You now briefly mention trade-offs with ecosystem services in the introduction, but this points remains insufficiently unaddressed in the discussion. You only discuss this in one sentence 'Commercial and conservation forests provide different but vitally important ecosystem services'. So what kind of trade-offs are relevant in the context of UK forests, when comparing conservation/commercial? I know word limits are tight, but in my opinion, this is a crucial points and links to my final two points.
6. I think the final message/recommendation still needs to sharpened. The UK Committee on Climate Change has proposed this figure of 30,000ha/yr for afforestation, which you advocate should be fully met by commercial forests (?). But at the same time you state that 'Commercial and conservation forests provide different but vitally important ecosystem services and should be considered complementary and not conflicting land uses. Rather than limiting the area of commercial forests in favour of semi-natural forests⁸ 398, more emphasis should be placed on ensuring good forest management that delivers both wood products and other ecosystem services'. Are you suggesting that conservation forests should not be considered as an option within this 30,000ha budget? Or are you suggesting that within this 30,000ha figure, a portion should be dedicated to commercial forestry? This links to point 5 about trade-offs with ecosystem services other than climate regulation.
7. You then suggest, that land can be retrieved, for the purpose of afforestation, by means of sustainable intensification and curtailed demand for livestock products. You also mention climate-smart forestry. How are these options related? I would recommend you elaborate on this a bit more, as now such considerations seem a bit vague and generic. Sometimes, the references you provide for such statements are a bit generic, e.g. about sustainable intensification and curtailed demand for livestock products. I know space is limited, but these final recommendations are possibly the most important part for readers. Once your key message is clear, please adjust title and abstract accordingly.

Reviewer #3 (Remarks to the Author):

Dear Authors, dear Editor,

Thank you for the opportunity to review the manuscript "Commercial forestry delivers effective climate change mitigation under multiple decarbonisation pathways", which I have read with great interest. The paper aims to generate new evidence on the climate change mitigation efficacy resulting from the establishment of commercial forests versus conservation forests in the UK. The paper concludes that commercial Sitka spruce forests achieve up to 269% more GHG mitigation than semi-natural broadleaf conservation forests, and 3-17% more mitigation than non-harvested conifer conservation forests. I have a few overall comments as well as a few detailed suggestions/questions that would need to be addressed prior to possible publication.

Overall comments:

- In my view the main contributions of the paper relate to 1. pointing out that active forest management on afforested sites can under certain circumstances provide net climate benefits in a time horizon of a century, despite that this seems highly unlikely in some other contexts such as

Northern boreal forests (or at least in the context of clear felled and replanted sites, if not afforested and managed), and that 2. the deployment of BECCS would not necessarily imply additional mitigation, but rather maintain the current credits from bioenergy substitution impacts. It is also good that the paper assumes that trees from afforestation sites are likely to be used commercially, which alters their climate impacts across different time horizons.

- However, it does not seem well justified to compare afforestation in sites with no competing land uses to conservation of existing semi-natural forests, as they are not competing strategies as the authors themselves mention. A more relevant comparison would be for a management decision (harvest or no harvest, or which type of harvest and for which purpose) for a single existing site. Now there is a sense of comparing apples and pears, as it is not a surprising result that newly established young forests imply a larger net carbon sink than conservation or management of an old forest: Additional afforestation is naturally more beneficial, if you frame it like this.

- Related to this, the results are somewhat difficult to follow, as there is no counterfactual analysis. It would be much easier to interpret the results, if the net GHG emissions were first quantified for all scenarios and then reported in relation to a baseline (counterfactual) scenario showing the marginal impact of the scenarios – in comparable decision situations.

- Overall, the methods seem appropriate and the illustrations are helpful. However, I am quite concerned on the assumptions for estimating substitution impacts. It is very odd that the material substitution impacts for the entire scope of wood uses are calculated based on a single building element (sawnwood substituting masonry walls), despite that a material flow for other product groups has been compiled as well. In previous literature (e.g., Werner et al. 2007, Smyth et al. 2017, Geng et al. 2019), the very minimum level has been to model an entire building with the same functionality (one more wood intensive and the other less wood intensive). The substitution credits for other markets than construction and energy are indeed uncertain as you note, but this is not a good enough reason to leave them unquantified. Instead, you should systematically consider all major market segments and perform sensitivity analysis on the most uncertain cases. Otherwise, this level of analysis yields very little in terms of scientific contribution or practical relevance, and there is not enough evidence to draw the conclusion found on lines 333-335 on the range of impacts of assuming different market structures. Besides current products, emerging wood products for e.g. textiles and packaging have been completely ignored despite a time horizon of a century and despite correctly accounting for the targeted decarbonisation of the energy sector and competing industries (although it is also not surprising that the decarbonisation seems to have little impact on the results, as you assume to substitute fossil fuels also under these circumstances). Moreover, some of the more detailed assumptions remain unclear, which I list below.

Detailed comments:

- Substitution impact estimates: Do you consider the possibility that UK sawnwood would be exported to other regions with different wood use structure? How realistic are the cascade scenarios in practice? Have you allocated substitution credits to the energy produced in pulp and saw mills (this should not be the case to avoid double counting with forest carbon sinks, as the substitution credits derive mostly from setting bioenergy as carbon neutral in the energy sector, see e.g. Rüter et al. 2016)? In relation to this, have you excluded biogenic emissions from the substitution factors? What is the denominator for the substitution factors (are the substitution factors calculated per harvested wood product or roundwood equivalent)?

- Terminology: Rather than commercial forests vs. conservation, IPCC and UNFCCC processes typically refer to sustainable forest management vs. conservation. Consider explicitly mentioning the term SFM.

Refs:

Geng, A., Chen, J., Yang, H., 2019. Assessing the greenhouse gas mitigation potential of harvested wood products substitution in China. *Environ. Sci. Technol.* 53, 1732–1740.

Rüter, S., Werner, F., Forsell, N., Prins, C., Vial, E., Levet, A.-L., 2016. *ClimWood2030-Climate benefits of material substitution by forest biomass and harvested wood products: Perspective 2030. Final report. Thünen Report.*

Smyth, C., Rampley, G., Lemprière, T.C., Schwab, O., Kurz, W.A., 2017. Estimating product and energy substitution benefits in national-scale mitigation analyses for Canada. *Gcb Bioenergy* 9, 1071–1084.

Werner, F., Taverna, R., Hofer, P., Richter, K., 2006. Greenhouse gas dynamics of an increased use of wood in buildings in Switzerland. *Clim. Change* 74, 319–347.

RESPONSE TO REVIEWER COMMENTS

Reviewer #1 (Remarks to the Author):

I think the authors still need to sharpen the key message of this manuscript, as the entire exercise is to provide insights for national decision-making.

Thank you for identifying a lack of clarity in our key messages. We have made a number of amendments to address this and strengthen our conclusions and messages. These are outlined in our responses to specific points, below.

Please consider the following points.

1. Should the title include 'in the UK'? Especially in consideration of what pointed out by myself and R2 in the previous revision round.

We feel that the findings and validity of the paper apply not just to the UK but to temperate regions more broadly, since many temperate environments share the key characteristics of the UK that have influenced the results of our study. These include the substrate and climate conditions of sites available for plantation forestry, plantation forest management, and the characteristics of forest product value chains. The introduction of sensitivity analyses testing product substitution also helps to broaden the scope of relevance beyond the UK, to reflect a wider range of possible timber value chains. However, it is true that we have used a UK context to define and test the effects of a possible national scale planting strategy, and we now emphasise in the Discussion section that our results pertain to regions where tree growth is relatively fast. So, we have left "UK" out of the title, but have subtly modified it to more precisely convey our focus with an implied caveat: "Commercial *afforestation can* deliver effective climate change mitigation...." (i.e. under certain conditions)

2. Line 44 'recognised in mitigation strategies' -> what mitigation strategies?

Thank you for noting this ambiguity. To clarify we now explicitly refer throughout the paper to 'GHG mitigation'

3. Line 48 'replaced' -> I think this should be 'replace' –

In the sentence "For slow-growing Boreal forests, carbon payback from bioenergy harvest has been estimated at 190-340 years, and is highly sensitive to the type of fossil fuel replaced in the future¹⁶" we believe that "replaced" is grammatically correct. We do not think it is correct to change it to "replace".

4. Could you quickly explain, in the intro and discussion, how the number 30,000ha/yr was selected? I understand that it is a target/recommendation suggested by the UK Committee on Climate Change, but this is not so evident to any reader

Thank you for noting this lack of explanation. We have added in this information for clarity, 'to contextualise the results in a national planting strategy, the UK's proposed strategy of 30,000 ha/yr from 2020 to 2050 was modelled.' This strategy is outlined in the UK Climate Change Committee (2019) report on Net Zero GHG emissions.

5. You now briefly mention trade-offs with ecosystem services in the introduction, but this points

remains insufficiently unaddressed in the discussion. You only discuss this in one sentence 'Commercial and conservation forests provide different but vitally important ecosystem services'. So what kind of trade-offs are relevant in the context of UK forests, when comparing conservation/commercial? I know word limits are tight, but in my opinion, this is a crucial points and links to my final two points.

We agree that the trade-offs in delivery of different ecosystem services are a key factor in selecting forestry options, and that this issue would be a worthy subject for further rigorous analysis in a new paper. We can see the potential for our life cycle assessment of GHG mitigation to be an important component for such a new analysis. The complexity of this analysis will be increased by the issue that the best solution for delivery of the range of ecosystem services required by society is likely to be achieved by a portfolio of different forest types at a landscape scale. Given this, we believe that presentation of new findings on this issue should only be done when they can be evidenced by this new analysis. Instead we see the most valuable role of the present paper being to present the most rigorous possible analysis of the role of alternative forestry options for GHG mitigation, which will provide a valuable component of evidence into future studies of trade-offs in the delivery of different ecosystem services. We now emphasise in the paper that this is the specific objective of our study, see sentences: "Although prioritising GHG mitigation in afforestation strategies will involve complex trade-offs with other ecosystem services¹⁷⁻²⁰, such as biodiversity conservation, the purpose of this paper is not to explore these. Rather, it aims to generate robust evidence on the efficacy of alternative afforestation options for the primary strategic aim of mitigating GHG emissions.". But we do also model a scenario of mixed commercial and conservation forest planting to at least partially demonstrate possible trade-offs between more conservation-oriented approaches and GHG mitigation maximisation, and we do elaborate considerably more in the discussion, e.g. penultimate paragraph, on ecosystem services trade-offs.

6. I think the final message/recommendation still needs to sharpened. The UK Committee on Climate Change has proposed this figure of 30,000ha/yr for afforestation, which you advocate should be fully met by commercial forests (?). But at the same time you state that 'Commercial and conservation forests provide different but vitally important ecosystem services and should be considered complementary and not conflicting land uses. Rather than limiting the area of commercial forests in favour of semi-natural forests^{8 398}, more emphasis should be placed on ensuring good forest management that delivers both wood products and other ecosystem services'. Are you suggesting that conservation forests should not be considered as an option within this 30,000ha budget? Or are you suggesting that within this 30,000ha figure, a portion should be dedicated to commercial forestry? This links to point 5 about trade-offs with ecosystem services other than climate regulation.

Thank you, we have made a number of edits to clarify this key component of our message. Our key finding is that fast growing species are critical for GHG mitigation, and commercial forestry is a robust GHG mitigation strategy regardless of wider societal decarbonisation pathways and wood uses. We acknowledge the important role of conservation forests in delivering other ecosystem services. However, we do not believe that meeting this objective should lead to a reduction in the commitment inherent to the UK Committee on Climate Change recommendation of 30,000 ha/yr of afforestation being targeted at the type of forest that will make the biggest contribution to GHG mitigation, i.e. fast-growing commercial forest. To reflect your important point, we now conclude that the 30,000 ha yr⁻¹ planting target should be increased to allow for a mix of commercial and

conservation forestry. This is an important elaboration, and so we are grateful to the reviewer for raising this point.

7. You then suggest, that land can be retrieved, for the purpose of afforestation, by means of sustainable intensification and curtailed demand for livestock products. You also mention climate-smart forestry. How are these options related? I would recommend you elaborate on this a bit more, as now such considerations seem a bit vague and generic. Sometimes, the references you provide for such statements are a bit generic, e.g. about sustainable intensification and curtailed demand for livestock products. I know space is limited, but these final recommendations are possibly the most important part for readers. Once your key message is clear, please adjust title and abstract accordingly.

We do not see a direct relationship between retrieving land (making land available for afforestation, by conversion from agricultural land use) and sustainable forest management (to which the term “climate-smart forestry” is linked). Nonetheless, we have added new text to better explain each of these two points in the discussion.

Land being freed by sustainable intensification and curtailed demand for livestock products is relevant because it explains why we do not account for avoided emissions from reduced livestock product in our consequential LCA. This assumption follows from recent reports and papers highlighting the high potential for land sparing via sustainable intensification and diet change in the UK. We now elaborate this point more clearly in the paper.

We introduced the point about sustainable forest management to highlight that commercial forestry does not need to mean single-species systems with low biodiversity value.

Reviewer #3 (Remarks to the Author):

Dear Authors, dear Editor,

Thank you for the opportunity to review the manuscript “Commercial forestry delivers effective climate change mitigation under multiple decarbonisation pathways”, which I have read with great interest. The paper aims to generate new evidence on the climate change mitigation efficacy resulting from the establishment of commercial forests versus conservation forests in the UK. The paper concludes that commercial Sitka spruce forests achieve up to 269% more GHG mitigation than semi-natural broadleaf conservation forests, and 3-17% more mitigation than non-harvested conifer conservation forests. I have a few overall comments as well as a few detailed suggestions/questions that would need to be addressed prior to possible publication.

Overall comments:

- In my view the main contributions of the paper relate to 1. pointing out that active forest management on afforested sites can under certain circumstances provide net climate benefits in a time horizon of a century, despite that this seems highly unlikely in some other contexts such as Northern boreal forests (or at least in the context of clear felled and replanted sites, if not afforested and managed), and that 2. the deployment of BECCS would not necessarily imply additional mitigation, but rather maintain the current credits from bioenergy substitution impacts. It is also good that the paper assumes that trees from afforestation sites are likely to be used commercially, which alters their climate impacts across different time horizons.

- However, it does not seem well justified to compare afforestation in sites with no competing land uses to conservation of existing semi-natural forests, as they are not competing strategies as the authors themselves mention. A more relevant comparison would be for a management decision (harvest or no harvest, or which type of harvest and for which purpose) for a single existing site. Now there is a sense of comparing apples and pears, as it is not a surprising result that newly established young forests imply a larger net carbon sink than conservation or management of an old forest: Additional afforestation is naturally more beneficial, if you frame it like this.

Thank you for highlighting a lack of clarity in our description of the baseline scenario. We have taken this opportunity to edit our introduction to ensure the reader can see that we are referring *only* to new woodland creation in this study. We adopted the term “conservation forest” to conform to internationally accepted terminology for contrasting forest types⁷ but in the light of the reviewer’s comment we realise that this did create considerable confusion in that it was taken to mean the management option of conserving *existing* semi-natural forest. Hence, we clarify here, and have better clarified in the manuscript that the options analysed in the paper are *all* alternative forms of *newly planted* afforestation. I.e. when comparing conservation (unharvested) woodland with commercial (harvested) woodland they are alternative forms of newly planted afforestation system. Hence, we are comparing equivalent land use change options that represent current decision making by landowners and policy makers in the context of current UK government targets for afforestation (forest creation). Thus, we are glad to report that our study does, indeed report on management decision (harvest or no harvest), or alternative forms of unharvested forest, for a single site, as advocated by the reviewer.

- Related to this, the results are somewhat difficult to follow, as there is no counterfactual analysis. It would be much easier to interpret the results, if the net GHG emissions were first quantified for all scenarios and then reported in relation to a baseline (counterfactual) scenario showing the marginal impact of the scenarios – in comparable decision situations.

As mentioned in response to comment 7 from Reviewer 1, this study assumes afforestation on land that has become available due to political and social change leading to reduced livestock production. Hence, in year zero we account for land use change from degraded grassland to forest... and we do not credit avoided emissions from avoided livestock production. The global warming potential impact of the land use change from grassland to forest is accounted for in the ‘terrestrial carbon storage’ category in the LCA of each afforestation scenario. We do not believe that, for example, natural regeneration of this grassland over 100 years is a realistic counterfactual due to commercial requirements of land owners coupled with extensive afforestation targets. Hence, the counterfactual analyses are the alternative afforestation scenarios we present. Even if we were to select a different counterfactual, because we are considering alternative afforestation options on a given area of land, the counterfactual GHG balance would be equal across all forest type scenarios – and would thus not influence comparative GHG mitigation efficacy results across forest types.

- Overall, the methods seem appropriate and the illustrations are helpful. However, I am quite concerned on the assumptions for estimating substitution impacts. It is very odd that the material substitution impacts for the entire scope of wood uses are calculated based on a single building element (sawnwood substituting masonry walls), despite that a material flow for other product groups has been compiled as well.

Thank you for raising this important point. We recognise that product substitution is an area in which there is great uncertainty, especially in a future context. We feel that our baseline approach is conservative but agree that this area would benefit from some further analysis. Hence, we have added new sensitivity analyses on some of the areas of greatest uncertainty to establish to what degree they affect the outcomes of the study. Our LCA is consequential and the base case product substitution represents a UK scenario where the additional structural timber from this new forest is supplying a shift in house design from concrete block to timber frame (i.e. concrete is the marginal product).

Details of the new sensitivity analyses are provided in supplementary information and outlined in the main text. They are also summarised below.

In our baseline product substitution assumptions, we only accounted for product substitution of carcassing. We have not accounted for potential substitution of any other products because we conservatively assume that other products (e.g. fence posts, pallets, wood boards) are predominantly sourced from wood in any case. In our sensitivity analyses we present two further product substitution analyses – one with reduced substitution credit and one with increased substitution credit. In our baseline, we assume that 100% of the carcassing is used to substitute concrete block walls due to a social/political shift leading to the construction of houses using timber-frames instead of concrete blocks. In the first test, in order to specifically address the recommendation of the reviewer, we apply reduced substitution credit by making a very pessimistic assumption that only 50% of the carcassing substitutes concrete block, with the remainder achieving no substitution credit. In a second sensitivity test, we estimate the effects of introducing glulam substitution for structural steel and viscose fibre substitution for cotton or PET fibre.

In addition to these product substitution tests we also decided to test the sensitivity of results to the timing of carbon capture and storage (CCS) technology deployment because we believe this is another area of high uncertainty. Therefore, in a third sensitivity test we quantified the impact of zero CCS deployment (to both bioenergy and fossil fuel energy systems) during the study timeframe. 100-yr GHG mitigation proved remarkably robust to these sensitivity tests, which therefore reinforce one of our core conclusions.

In previous literature (e.g., Werner et al. 2007, Smyth et al. 2017, Geng et al. 2019), the very minimum level has been to model an entire building with the same functionality (one more wood intensive and the other less wood intensive). The substitution credits for other markets than construction and energy are indeed uncertain as you note, but this is not a good enough reason to leave them unquantified. Instead, you should systematically consider all major market segments and perform sensitivity analysis on the most uncertain cases. Otherwise, this level of analysis yields very little in terms of scientific contribution or practical relevance, and there is not enough evidence to draw the conclusion found on lines 333-335 on the range of impacts of assuming different market structures. Besides current products, emerging wood products for e.g. textiles and packaging have been completely ignored despite a time horizon of a century and despite correctly accounting for the targeted decarbonisation of the energy sector and competing industries (although it is also not surprising that the decarbonisation seems to have little impact on the results, as you assume to substitute fossil fuels also under these circumstances).

We carried out a thorough literature review to establish the rate of decarbonisation and also to identify/estimate the projected marginal energy mix during the 100-year period. The fossil fuel substitution that we used is based on what is deemed to be the *marginal* fuel, with the appropriate application of CCS technology, depending on the year of emissions occurring. This means that over

time although fossil fuels may still be the marginal fuel, the substitution credits approach zero as CCS technology applied to energy generation mitigates GHG emissions. In the cases where bioenergy is deemed to be the marginal fuel, there is no substitution credit.

As mentioned in our response to the reviewer's previous point above, in order to specifically address the recommendation of the reviewer, we carried out a new sensitivity analysis to estimate the effects of introducing viscose fibre substitution for PET fibre.

Moreover, some of the more detailed assumptions remain unclear, which I list below.

Detailed comments:

- Substitution impact estimates: Do you consider the possibility that UK sawnwood would be exported to other regions with different wood use structure?

The UK currently imports 98% of its sawn softwood timber needs and its consumption rate is rising. This justifies our assumption that afforestation will supply some of the UK's current and predicted increased demand. Given the UK's low current cover of productive forests relative to its large domestic timber demand it is implausible that there will be significant export of sawnwood unless there was a level of afforestation orders or magnitude above those of current government targets, as modelled in our study. We recognise this is not the case in other high timber producing temperate countries, however for these same market reasons very few temperate countries with current high rates of timber production have government targets for such a high percentage increase in forest cover as does the UK. Our interpretation of international as well as national policies for climate change mitigation is that there will be a general increase in emphasis on in-country production to meet growing domestic demands for sawnwood rather than this being met primarily by an increase in international trade.

How realistic are the cascade scenarios in practice?

The cascade scenarios are based on real UK forest product and sawmill data from Gresham¹ House and James Jones Sawmills², respectively, provided in correspondence. Recycling data is obtained from UK government annually published statistics³. Product retiral rates are calculated according to IPCC methodology^{4&5}.

Have you allocated substitution credits to the energy produced in pulp and saw mills (this should not be the case to avoid double counting with forest carbon sinks, as the substitution credits derive mostly from setting bioenergy as carbon neutral in the energy sector, see e.g. Rüter et al. 2016)? In relation to this, have you excluded biogenic emissions from the substitution factors?

Thank you for pointing this out. In fact we had used Ecoinvent data⁶ for sawmills based on (external) wood fuel input. Although use of this process did not include fossil fuel substitution credits, we did neglect to subtract the equivalent flow of sawdust and woodchips from downstream heat and energy generation linked with fossil fuel substitution. We have now corrected this error, which resulted in a 0.6-1% reduction in overall forest value chain mitigation.

Biogenic emissions are fully accounted for, but separately from substitution factors. Our model treats biogenic emissions as the deficit between C stored in the forest system and C stored in harvested wood products (HWP), and, in future years in some scenarios, CCS. HWP C storage is calculated according to IPCC methodology of HWP 'decay' i.e. product retiral. Therefore, biogenic C

emission to the atmosphere is implicitly attributed to all products that reach the end of their lives and are combusted (i.e. biomass fuel or incinerated waste) or that decompose (i.e. horticultural mulch⁸ or waste wood in landfill⁹) during the study timeframe. Fossil fuel substitution benefits of biomass fuel are calculated as described below in response to your next point.

What is the denominator for the substitution factors (are the substitution factors calculated per harvested wood product or roundwood equivalent)?

1kWh of useful biomass energy is deemed to substitute 1kWh of useful energy produced from the appropriate fossil fuel. The FF substitution factors for biomass fuel are therefore determined by the emissions produced per kWh of the relevant substituted fuel. Biomass kWh useful energy is calculated according to the mass, moisture content, associated lower heating value, and average conversion efficiency, for specific wood flows (see S2&S3, specifically sheets 'Energy&FFDisplacement' and 'material flow').

- Terminology: Rather than commercial forests vs. conservation, IPCC and UNFCCC processes typically refer to sustainable forest management vs. conservation. Consider explicitly mentioning the term SFM.

SFM is broad term that is applied to a wide range of forest types and management options. While it is generally related to the simultaneous delivery of a wide range of ecosystem services (see our response to reviewer 1 point 5), consider that the key component of SFM for the present study is its fundamental "sustained yield", i.e. that the harvesting of timber from a forest should be carried out at a rate that does not reduce the yield of timber in future rotations. This is a powerful indicator that the form of forest management is not diminishing the "natural capital" and "supporting services" category of ecosystem services of the forest (nutrient cycling, primary production and, increasingly emphasised in the literature, functional biodiversity). As such, SFM equates fully with the commercial forestry option analysed in our study, which (crucially) is based on the rate of production of commercial conifer species that can be sustained through successive rotations. For clarity, we have also introduced the additional descriptive terms harvested and unharvested, in order to improve the clarity of the distinction between the afforestation options that we are comparing.

References

1. Gresham House. UK forest production data for Gresham House managed assets, provided from personal correspondence.(2018).
2. DEFRA. ENV23 - UK statistics on waste - GOV.UK. (2018). Available at: <https://www.gov.uk/government/statistical-data-sets/env23-uk-waste-data-and-management>. (Accessed: 31st December 2019)
3. James Jones & Sons. Sawmill production data provided from personal correspondence. (2019).
4. IPCC. *2006 IPCC Guidelines for National Greenhouse Gas Inventories Volume 4 Chapter 4*. (2006).
5. IPCC. *2006 IPCC Guidelines for National Greenhouse Gas Inventories Volume 4 Chapter 2*. (2006).
6. Wernet, G. *et al.* The ecoinvent database version 3 (part I): overview and methodology. *Int. J. Life Cycle Assess.* **21**, 1218–1230 (2016).
7. FAO (2004) Global forest resources assessment update 2005, Terms and definitions (Final Version), Forest Resources Assessment WP 83, Rome, 2004

8. Brunn, S. *et al.* Application of processed organic municipal solid waste on agricultural land - a scenario analysis. *Environmental Modeling and Assessment*. **11**, 251-265 (2006).
9. IPCC. *2006 IPCC Guidelines for National Greenhouse Gas Inventories Volume 4 Chapter 3*. (2006).

REVIEWERS' COMMENTS

Reviewer #1 (Remarks to the Author):

I have no further comments.

Reviewer #3 (Remarks to the Author):

Many thanks for the revised manuscript and the excellent clarifications on the points raised. The major concerns seem to have been misunderstandings originating from the use of concepts, and the clarifications and the associated text revisions have addressed these points very well.

As the only remark, the extra sensitivity analyses on substitution impacts are much appreciated. However, for future reference, the assumptions on substitution impacts in this paper remain thin compared to state-of-the art understanding under this topic. Although the various drawbacks and uncertainties regarding substitution could have been acknowledged much more explicitly, this can be tolerated given the very extensive scope of the study and the fact that the extent of substitution impacts are not likely to alter the conclusions of the study.

RESPONSE TO REVIEWER COMMENTS

Reviewer #1 (Remarks to the Author):

I have no further comments.

Reviewer #3 (Remarks to the Author):

Many thanks for the revised manuscript and the excellent clarifications on the points raised. The major concerns seem to have been misunderstandings originating from the use of concepts, and the clarifications and the associated text revisions have addressed these points very well.

As the only remark, the extra sensitivity analyses on substitution impacts are much appreciated. However, for future reference, the assumptions on substitution impacts in this paper remain thin compared to state-of-the art understanding under this topic. Although the various drawbacks and uncertainties regarding substitution could have been acknowledged much more explicitly, this can be tolerated given the very extensive scope of the study and the fact that the extent of substitution impacts are not likely to alter the conclusions of the study.

Author response: We thank the reviewers for their thorough reviews and constructive comments. We appreciate Reviewer #3's remark about substitution uncertainty and agree fully with this. As the reviewer also concludes, we are confident that we have applied a suitably conservative approach to substitution credits for our default results and that the sensitivity analyses confirm our conclusions are robust to uncertainty in future substitution credits. Deeper exploration of this topic is certainly beyond the scope of this paper and would distract from the overarching conclusions that integrate substitution effects with terrestrial carbon storage, harvested wood product carbon storage, carbon capture & storage, and multi-product cascading wood chains.